# Pretreatment Method for DNA Barcoding to Analyze Gut Contents of Rotifers

**Hye-Ji Oh [1], Paul Henning Krogh [2], Hyun-Gi Jeong [3], Gea-Jae Joo [4], Ihn-Sil Kwak [5,6], Sun-Jin Hwang [1], Jeong-Soo Gim [4], Kwang-Hyeon Chang [1,*] and Hyunbin Jo [6,*]**

[1] Department of Environmental Science and Engineering, Kyung Hee University, Yongin 17104, Korea; ohg2090@naver.com (H.-J.O.); sjhwang@khu.ac.kr (S.-J.H.)

[2] Department of Bioscience, Aarhus University, Vejlsøvej 25, 8600 Silkeborg, Denmark; phk@bios.au.dk

[3] Nakdong River Environment Research Center, National Institute of Environmental Research, Goryeong 40103; jhgpl@korea.kr

[4] Department of Integrated Biological Science, Pusan National University, Busan 46241, Korea; gjjoo@pusan.ac.kr (G.-J.J.); kjs1@pusan.ac.kr (J.-S.G.)

[5] Faculty of Marine Technology, Chonnam National University, Yeosu 59626, Korea; inkwak@hotmail.com

[6] Fisheries Science Institute, Chonnam National University, Yeosu 59626, Korea

* Correspondence: chang38@khu.ac.kr (K.-H.C.); prozeva@jnu.ac.kr (H.J.);
  Tel.: +82-10-8620-4184 (K.-H.C.); +82-10-8807-7290 (H.J.)

**Abstract:** We designed an experiment to analyze the gut content of Rotifera based on DNA barcoding and tested it on *Asplanchna* sp. in order to ensure that the DNA extracted from the rotifer species is from the food sources within the gut. We selected ethanol fixation (60%) to minimize the inflow effects of treated chemicals, and commercial bleach (the final concentration of 2.5%, for 210 s) to eliminate the extracellular DNA without damage to the lorica. Rotifers have different lorica structures and thicknesses. Therefore, we chose a pretreatment method based on *Asplanchna* sp., which is known to have weak durability. When we used the determined method on a reservoir water sample, we confirmed that the DNA fragments of Chlorophyceae, Diatomea, Cyanobacteria, and Ciliophora were removed. Given this result, Diatomea and cyanobacteria, detected from *Asplanchna*, can be considered as gut contents. However, bacteria were not removed by bleach, thus there was still insufficient information. Since the results of applying commercial bleach to rotifer species confirmed that pretreatment worked effectively for some species of rotifers food sources, in further studies, it is believed to be applicable to the gut contents analysis of more diverse rotifers species and better DNA analysis techniques by supplementing more rigorous limitations.

**Keywords:** gut content of Rotifera; eliminate the extracellular DNA; commercial bleach; pretreatment

## 1. Introduction

It is important to understand the role and function of interactions in the microbial food web of aquatic ecosystems. The key biological interaction in the aquatic microbial food web is matter cycling mediated by predation, and predation often works as a regulating factor for energy pathways, as well as determining species composition [1]. In particular, rotifers are critical components linking microorganisms with larger predatory organisms such as crustaceans and fish within the grazing food chain: bacteria, heterotrophic nano-flagellates, rotifers/copepods/cladocerans, larval fish, mature fish [2,3]. Consequently, they function as a channel for the flux of organic matter within diverse organism assemblages organized in an intermediate position between the two different food webs, and transfer nutrients and energy from the microbial loop to higher trophic levels [4–6]. In addition, as the problem of eutrophication increases in aquatic ecosystems, the abundance of macrozooplankton

decreases and consequently the contribution of rotifers in energy flow of aquatic food web becomes greater [2]. As a result, rotifer-focused biological interactions, especially rotifer feeding behaviors in microbial food web, are receiving a great attention to understand not only the interrelated biological relationships but also the structure and function in aquatic food webs [7].

However, the comprehensive understanding of rotifers feeding characteristics has not been well-elucidated in comparison to their importance, because most previous studies were conducted at the lab-scale with limited environmental conditions over a short time period, limited to common and dominant species as the tested species, and therefore have not been verified in the field [8–12]. These limitations were attributed to the absence of adequate analytical methodologies applicable to field sites due to difficulties in culturing, handling and identification of both prey and predator (rotifers) which have small sizes (usually rotifers body size≤ 1000 μm; rotifers prey size spectrum<1–20 μm) [13]. In order to overcome the methodological limitations for the analysis of rotifer feeding behaviors, the introduction and application of appropriate techniques are required.

In recent years, genomic technologies have developed rapidly and been applied to ecological research. DNA barcoding techniques have increased the reliability of identifying specific taxonomic groups of organisms at both species and genus level [14], and environmental DNA techniques have enabled the detection of elusive species in various environments [15,16]. Genomic approaches have also been used to understand trophic ecology, particularly biological interaction, for both aquatic habitat environments and food webs by collecting information from food material found in gut contents and the excrement of various organisms and this helps to overcome the existing limitations of food source analyses, which were usually based on visual analysis [17–20].

So far, however, the microscopic and DNA identification of food remains in the gut contents have been limited to large-size organisms such as fish and benthic macroinvertebrates as gut contents extraction is difficult to perform. In the case of zooplankton, crustaceans, with relatively large body size (usually larger than 1 mm) and a hard exoskeleton structure, such as a carapace, which covers the digestive organs, have been the main target for food source analysis. Their morphological characteristics allow physical and chemical treatments, as well as dissection to extract gut contents, avoiding DNA fragments from microorganisms attached to their bodies and DNA from the predator itself. In practice, diets analyses of copepods (small crustaceans) using the DNA-based methods were conducted in both freshwater and ocean ecosystems [21,22]. On the other hand, since small rotifers (usually < 0.5 mm) are relatively soft-bodied, it is difficult to apply similar physical and chemical treatments as for other zooplankton, and there are no proper methodologies and sufficient information of rotifer food sources as results [23]. For a wide range of applications of DNA technology in food source identification, it is necessary to develop a method for separating gut content items from an object by minimizing other possible DNA contaminants, no matter how small the target size is.

For applications of DNA technology to the identification of rotifers food sources, the most critical part of methodology is to distinguish the DNA in the rotifers gut contents from contamination sources that can be attached to the outside of the rotifers lorica and exist in the sample water (so-called 'extracellular DNA'). Since the detection of extracellular DNAs can cause confusion in the interpretation of the results of the rotifers gut contents analysis, treatment for eliminating them (so-called 'pretreatment') is necessary to obtain the more accurate results of rotifers gut contents analysis. However, unlike crustacean zooplankton, which have a solid carapace, the rotifer body is covered by a lorica, which is relatively softer than the carapace. In addition, lorica hardness is differs by species [24].

In this study, we focused on the establishment of an appropriate method for detecting the DNA of gut contents, which is applicable for soft-bodied rotifers. In this analysis procedure, it is important to eliminate the cells and DNA fragments of microorganisms attached to rotifers in order to extract and analyze only those food sources included in the gut to eliminate extra-cellular DNA contamination. Therefore, we selected chemicals for eliminating different types of extracellular DNA and tested their effects on the lorica of rotifers under different concentration treatments to find the most effective

concentration and time for both the preservation of the rotifers and removal of different types of extracellular DNA. Following this, we tested the applicability of gut content analysis to rotifers using DNA technology by verifying whether the DNA fragments of rotifer food sources were eliminated or not when the prescribed treatment method was used.

## 2. Materials and Methods

### 2.1. Selection of Chemicals for Eliminating Extracellular DNAs

In order to select the appropriate chemicals to remove the extracellular DNA fragments in detecting DNA of rotifers gut contents, we reviewed the different treatment methods and their procedures found in the literature (Table 1, Table A1).

**Table 1.** Summary of previous treatment processes for decontamination in DNA analyses.

| Target | Treatments and Conditions | Ref. |
|---|---|---|
| Contaminated DNA for extinguishing the template activity | Incubation of DNA with a psoralen, 8-methoxypsoralen in the dark for 30 min to overnight and subsequent irradiation of UV (365 nm) for 1 hr. | [25] |
| Bone for removal of contamination | Washed in sterile distilled water, followed by 10% bleach * | [26] |
| Teeth for destroying any contaminating DNA on the surface | Soaked in hydrogen peroxide (3–30%) for 10–30 min, rinsed with distilled water, rinsed thoroughly with 10% bleach *, rinsed again with distilled water and UV irradiated for 10 min | |
| Teeth and cortical bone pieces to prevent extraneous contaminations (dirt, carbonate deposits, acid residues) | Soaked for 10 min in 15 % HCL, for 10 min in 70 % ethanol and rinsed in sterile double-distilled water for 30 min | [27] |
| Tooth for prevention of contamination | Soaked for ~ 10 min in 10% bleach * and then rinsed with 70 % ethanol | [28] |
| Skeletal material (e.g., powdered bone) for reducing DNA contamination | - Immersing in 20 % bleach * for 2 min followed by extensive ddH$_2$O washing - 2 days treatment with 0.5 M EDTA at 55 °C | [29] |
| Bone fragments for elimination of any minor surface contamination | 10 min on each side with UV light (254 nm) and soaked for approximately 5 min in a 5 % bleach solution* (in some cases) | [30] |
| Teeth for removal of dirt and other contaminants | Treated with 30 % acetic acid, rinsed with ultrapure water, immersed for 10 min in 10 % sodium hypochlorite* and exposed to UV light (254 nm) for at least 10 min on each side | [31] |
| Environmental samples for removal of free extracellular DNA | Added Ethidium or Propidium Monoazied (EMA or PMA) following a conventional procedure in accordance with the manufacturer's protocol (PowerSoil DNA extraction kit (MoBio Lab, Inc.) | [32] |
| External DNA contamination of arthropod gut-content | 40 min of end-over-end rotation in 2.5 % commercial bleach * | [33] |

\* Underlined treatment and procedure: the case of the application of bleach for decontamination

Most previous published treatment methods were not suitable for the selective elimination of external DNA, which is required for gut content analysis of rotifers, due to their soft lorica. Therefore, we selected ethanol and commercial bleach to remove extracellular DNA from the rotifers, while maintaining the internal gut content DNA composition inside the rotifers. Ethanol (Ethyl alcohol; CAS No. 64-17-5) was used for preservation and sterilization of the raw water sample. Accordingly, the raw water sample was fixed with ethanol to a final concentration of 60% (typical concentrations for disinfection and sterilization: 60–95% [34]). Preserving rotifers with ethanol would limit the exposure to damage of the rotifer lorica by the action of commercial bleach, which has previously been used for DNA elimination and extraction of gut contents in zooplankton, only externally [33,35]. Yuhan-Clorox

(Yuhanrox regular) (Yuhan Co, Ltd., Korea) composed of 4%–6% NaClO (CAS No. 7681-52-9) and 0.1%–0.5 % NaOH (CAS No. 1310-73-2) was used for the chemical wash treatment.

### 2.2. Responses of Rotifers Lorica to Bleach

To find a suitable treatment time and concentration of commercial bleach for extracellular DNA removal without affecting rotifer lorica and gut contents, we measured the response time of different rotifer species, which were collected from a water reservoir, to different exposure concentrations. We minimized contamination by separating each sample, using bleach sterilized gloves, and instruments sterilized by autoclave and ethanol. As testing the response time and range of commercial bleach concentrations requires multiple individuals of each rotifer species, we targeted large species in which at least three individuals can be gathered by sorting, *Brachionus forficula*, *Keratella* sp., *Trichocerca* sp., *Polyarthra* sp., and *Asplanchna* sp. having variable lorica characteristics; from species having soft lorica (e.g., *Asplanchna* sp.) to hard lorica (e.g., *Keratella* sp. and *Polyarthra* sp.) [36]. Each rotifer species was treated using commercial bleach with final concentrations of 20%, 10%, 5%, and 2.5%. We measured the time until lorica disintegration of the three individuals per rotifer species through microscopic inspection (OLYMPUS CKX41). These time results were used as baselines to determine the concentration and time of removing extracellular DNA without damaging the rotifer individuals (Figure 1).

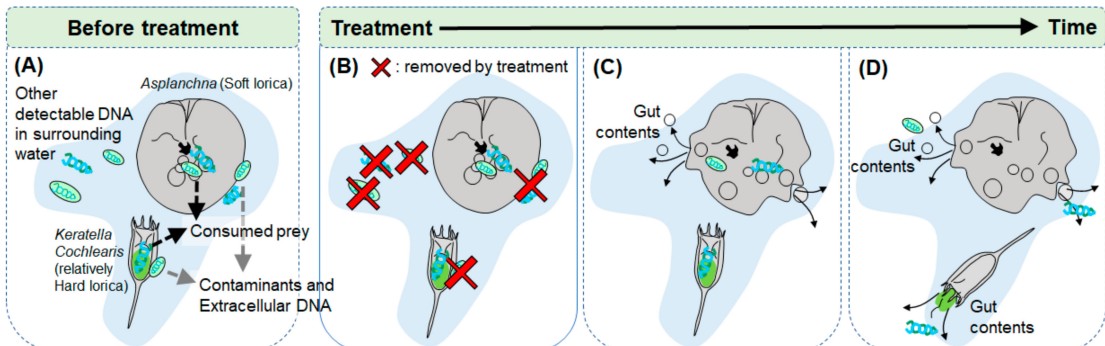

**Figure 1.** Diagram of necessity of proper pretreatment process in rotifers gut contents analysis; (**A**) Without treatment for removing contaminants and other detectable DNA including extracellular DNA, there is possibility to be confused that detected DNA is from the rotifers gut contents or not, (**B**) Through proper treatment, contaminants and other detectable DNA can be removed without damaging the rotifer individuals, (**C**) and (**D**) Rotifer individuals can be damaged and their gut contents can be overflowed by excessive treatment according to their lorica characteristics.

### 2.3. Application and Effectiveness Verification of Set Treatment Concentration and Time

We collected water samples from a eutrophic reservoir (Shin-gal reservoir, Korea; N 37.241536, E 127.0929190) in fall (4th November 2018). Rotifers dominated the zooplankton community of the reservoir during this season. We repeatedly filtered 10 L of surface water (n = 20) into a 60 μm mesh sized zooplankton net and obtained a 1 L filtered water sample. From the collected sample, all organisms were removed by hand using a microscope (OLYMPUS CKX41) and 0.5 mL of subsample was extracted to micro-tubes. For every sample taken, we made a negative control to prevent cross contamination.

For the application and effective verification of a set treatment concentration and time on the DNA fragments of rotifer food sources (Chlorophyceae, Diatomea, Cyanobacteria, Bacteria, Ciliophora, and Heterotrophic nanoflagellates) [9], we compared both treated and non-treated samples with 0.5 mL filtered water samples. In the case of treated sample, after set treatment time, we poured and filtered the sample immediately by washing with distilled water to prevent further effects. Non-treated sample was also filtered in order to proceed with the same DNA extraction process as the treated sample.

To confirm detection of DNA in the rotifers gut without extracellular DNA using the suggested treatment method, we collected a rotifer species, *Asplanchna* sp., from reservoir water, and applied this method to the treatment. We sorted *Asplanchna* specimens from the water sample and transferred them to distilled water several times until pure rotifer individuals were collected without other visible particles, particularly phytoplankton cells. We then checked for removal of particles under the microscope and selected clean individuals without attached particle or microorganisms (one individual per a sample, 3 replicates). As with water samples, rotifers samples were filtered after pretreatment for extraction of their DNA. The 47mm diameter cellulose nitrate filters with a pore size of 0.45μm pore size (NC 45 ST, Whatman[TM]) were used to filter DNA fragments [37].

### 2.4. DNA Analysis Precedure

The DNA was isolated from the filter paper using DNeasy Blood & Tissue kit (Qiagen, Hilden, Germany) according to the manufacturer's instructions on a clean bench (Supplementary Materials). In order to reduce potential contamination during DNA extraction and the amplification process, we minimized contamination sources by separating each sample, using bleach sterilized gloves, and instruments sterilized by autoclave and ethanol.

A polymerase chain reaction (PCR) amplification was performed using AccuPower Hot start PCR PreMix (Bioneer, Korea) with genomic DNA and primers in a final volume of 20 μL. Primers were selected for specific detection of each potential prey community for rotifers. Phytoplankton and the components of the microbial food web (bacteria and protozoa) were considered as potential food sources for rotifers [6]. At this time, since we could not find a suitable primer to detect only heterotrophic nanoflagellates (HNF) specifically, we applied a universal primer for eukaryotes to HNF [38,39]. PCR conditions for each primer in a thermal cycler (Bio-rad, California, USA) were summarized in Table 2. PCR products were separated using 1.5% agarose gels (AccuPrep® PCR/Gel DNA Purification Kit (50 reactions) [K-3038]), and the appeared band from PCR products was extracted and sequenced in both directions by capillary sequencing at Bioneer Co. (Daejeon, Korea). Additionally, cloning was carried out using the pGEM-T easy vector (Promega, Madison, USA) to confirm direct sequencing results. Cloned plasmid DNA was isolated according to the alkaline-lysis method using Labopass Plasmid Miniprep kit (Cosmogenetech). Individually isolated plasmid DNA was then digested using the restriction enzyme EcoRI to confirm insertion. Positive clones for each sample were analyzed to species-specific sequences with SP6 primers using an automated 3730 DNA analyzer (Applied Biosystems, Foster City, USA).

Sequence alignment was performed using Clustal W 2.0 [40]. A BLASTn [41] search was performed to identify sequences with the best hits. In the GenBank nucleotide collection database, the organisms, which were included in the database search, were optimized for highly similar sequences by BLASTn, and selected by high identity (%).

**Table 2.** PCR primers used for detecting rotifer food sources.

| Food source | Primer | Sequence (5'-3') | Base Pair | Ref. |
|---|---|---|---|---|
| Chlorophyceae (18s rRNA) | ChloroF ChloroR | TGG CCT ATC TTG TTG GTC TGT GAA TCA ACC TGA CAA GGC AAC | 473 bp | [42] |
| | 94 °C, 3 min -> 35cycles [94 °C, 1 min -> <u>55 °C **</u>, 1 min -> 72 °C, 1 min] -> 72 °C, 10 min | | | |
| Diatomea (18s rRNA; V4) | M13F-D512for M13R-D978rev | TGT AAA ACG GCC AGT ATT CCA GCT CCA ATA GCG CAG GAA ACA GCT ATG ACG ACT ACG ATG GTA TCT AAT C | 390–410 bp | [43] |
| | 94 °C, 2 min ->5cycles [94 °C, 45s -> <u>53 °C **</u>, 45s -> 72 °C, 1 min] ->35cycles [94 °C, 45 s -> <u>51 °C **</u>, 45s -> 72 °C, 1 min] -> 72 °C, 10 min | | | |

**Table 2.** *Cont.*

| Food source | Primer | | Sequence (5'-3') | Base Pair | Ref. |
|---|---|---|---|---|---|
| Cyanobacteria (16S rRNA; ITS *) | 16S27F | | AGA GTT TGA TCC TGG CTC AG | 422 bp | [44] |
| | 23S30R | | CTT CGC CTC TGT GTG CCT AGG T | | |
| | 94 °C, 5min -> 10cycles [94 °C, 45s -> <u>57 °C</u> **, 45s -> 68 °C, 2 min] -> 25cycles [92 °C, 45s -> <u>54 °C</u> **, 45s -> 68 °C, 2 min] -> 68 °C, 7 min | | | | |
| Bacteria (16S rDNA; nearly full-length) | Forward | | GAG TTG GAT CCT GGC TCA G | About 2000 bp | [45,46] |
| | Reverse | | AAG GAG GGG ATC CAG CC | | |
| | 95 °C, 3 min -> 35cycles [94 °C, 1 min -> <u>60 °C</u> **, 1 min -> 72 °C, 2 min] -> 72 °C, 3 min | | | | |
| Ciliophora (18S rRNA) | Cil F | | TGG TAG TGT ATT GGA CWA CCA | 600–670 bp | [47] |
| | Cil R- | 1 | TCT GAT CGT CTT TGA TCC CTT | | |
| | | 2 | TCT RAT CGT CTT TGA TCC CCT A | | |
| | | 3 | TCT GAT TGT CTT TGA TCC CCT | | |
| | 95 °C, 5 min -> 35cylces [94 °C, 45 s -> <u>55 °C</u> **, 1min -> 72 °C, 1 min] -> 72°C, 10min | | | | |
| Heterotrophic nanoflagellates (18s rRNA) | EukA *** | | AAC CTG GTT GAT CCT GCC AGT | 800–900 bp | [38,39] |
| | EukB *** | | TGA TCC TTC TGC AGG TTC ACC TAC | | |
| | 95 °C, 2 min -> 35cycles [95°C, 30s -> <u>55 °C</u> **, 30s -> 72 °C, 2 min] -> 72 °C, 7 min | | | | |

* Internal transcribed spacer; ** Underlined temperature: annealing temperature of each primer; *** EukA and EukB are universal primer for eukaryote.

## 3. Results

### 3.1. Responses of Rotifers Lorica to Commercial Bleach Treatment

After treatment with commercial bleach at final concentrations of 20%, 10%, 5%, and 2.5%, the time before the loss of each rotifer's contents by lorica disintegration (five rotifers species tested, n=3) was measured. Every tested rotifer species, *Brachionus forficula*, *Keratella* sp., *Trichocerca* sp., *Polyarthra* sp., and *Asplanchna* sp. tended to have shorter times for tolerating treatment as the final concentration of commercial bleach increased. In particular, *Asplanchna* sp. having the weakest lorica showed the shortest time among tested rotifers regardless of treatment concentration. The lorica of *Asplanchna* disintegrated between 35 s and 240 s following exposure to different treatment solutions of various concentration, and its body contents including the gut contents were released from the body. When *Asplanchna* was treated with 2.5% diluted commercial bleach, although it was observed to withstand up to 300 s of exposure, its lorica began to suffer disintegration after 240 s. Therefore, for preservation of its gut contents, the treatment time should be considered as less than 240 s. Other tested rotifer species also showed different duration times for lorica survival against treatment solution depending on its concentration. However, regardless of their lorica thickness and structure, they showed a range of endurance time from 300 to 450 s at 2.5% of commercial bleach (Table 3, Figure 2).

**Table 3.** The responses of rotifers lorica to commercial bleach treatment; the minimum time (s) before the loss of the rotifers contents by lorica disintegration of each rotifers species.

| Concentration (Final) and Duration Time | Rotifer Species | | | | |
|---|---|---|---|---|---|
| | *Brachionus Forficula* | *Keratella* sp. | *Trichocerca* sp. | *Polyarthra* sp. | *Asplanchna* sp. |
| 20% | 60 s | 90 s | 120 s | 60 s | 35 s |
| 10% | 210 s | 180 s | 150 s | 90 s | 45 s |
| 5% | 300 s | 240 s | 300 s | 210 s | 120 s |
| 2.5% | 450 s | 300 s | 450 s | 300 s | 240 s |

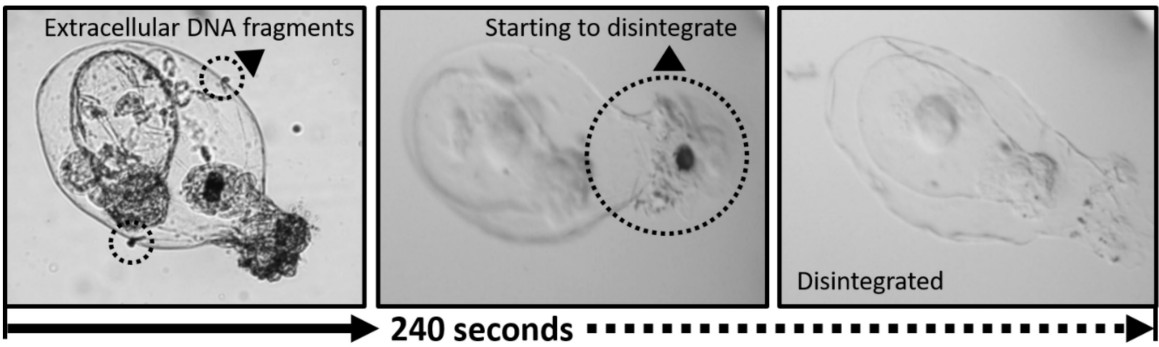

**Figure 2.** An example of a disintegration process; response of *Asplanchna* sp. lorica to commercial bleach treatment (2.5 %).

To prevent loss of gut contents during pretreatment for extracellular DNA by disintegration of rotifers lorica, we should establish the conditions (concentration and time of chemical) under which extracellular DNA can be removed, and keep the rotifer lorica undamaged. Since high concentration treatments allowed very limited time available for affecting the elimination of extracellular DNA, we decided to use the lowest concentration of commercial bleach for the longest time on samples in order to minimize damage while maximizing external DNA removal. Therefore, based on the response time of *Asplanchna* sp. lorica to the lowest concentration of commercial bleach treatment and consequent its shortest duration time examined by the experiment, rotifer specimens for extracting gut contents DNA were treated with 2.5% diluted commercial bleach for 210 s (Table 3). We observed each treatment process through a microscope, and confirmed that the gut contents of rotifers were likely to be released from the body when their lorica began to disintegrate. Therefore, we judged that it would be appropriate to use commercial bleach for removal of the extracellular DNA up to 30 s before the time when rotifer loricas begin disintegrating. In addition, to maximize treatment time while minimizing internal effects of the treatment by fixing rotifers, we determined that preservation with 60% ethanol soon after the samples are collected and treatment of 2.5% diluted commercial bleach for 210 s was the most effective pretreatment (Figure 3). We, therefore, selected *Asplanchna* sp. for our further experiment. In addition, *Asplanchna sp.* have a typical omnivore feeding behavior according to Chang et al. (2010) [48]. Therefore, it is an ideal experiment creature for this study.

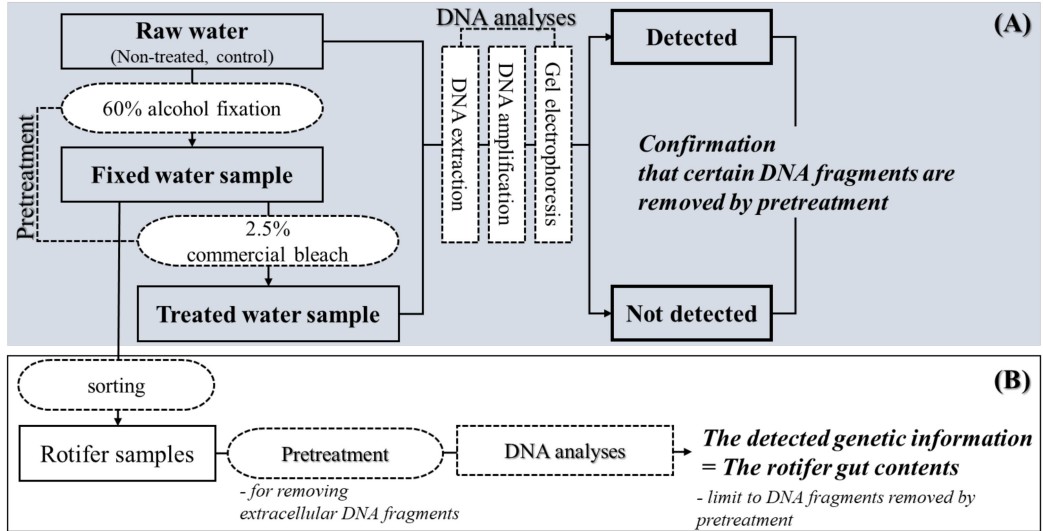

**Figure 3.** Experimental designs. (**A**) Verifying applicability of pretreatment, (**B**) Application of pretreatment to rotifers.

### 3.2. Application and Effectiveness Verification of Set Pretreatment Concentration and Time

When the electrophoresis results of raw water from reservoir (non-treated water sample), treated water and treated rotifers samples were compared, they showed different bands in each gel. In the non-treated water sample, the primers used to detect various regions of genetic sequences were all amplified and detected as bands in the electrophoresis gel (Figure 4A–E,N). As a result of identifying the dominant signal information of sequences by the direct capillary sequencing method through BLASTn, all dominant signal identified in non-treated water samples were of Chlorophyceae, Diatomea, Cyanobacteria, Bacteria, and Ciliophora, which are known as common food sources of rotifers (Table 4; Non-treated water sample).

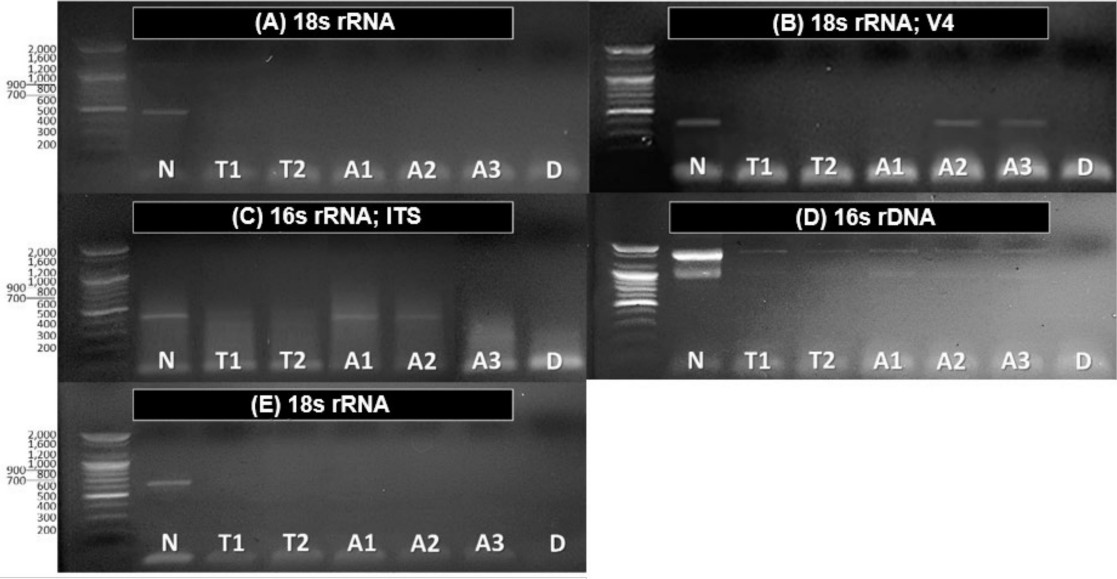

**Figure 4.** Electrophoresis detection results, (**A**) 18s rRNA for detecting Chlorophyceae; (**B**) 18s rRNA; V4; (**C**) 16s rRNA; ITS; (**D**) 16s rDNA; (**E**) 18s rRNA for detecting Ciliophora and HNF, heterotrophic nanoflagellates; N: non-treated water sample (raw water; control); T: treated water sample by ethanol (60%) and 2.5% commercial bleach solution; A: treated *Asplanchna* samples (n=3); D: distilled water (negative control); first lane of each gel: Ladder using 100-bp molecular marker.

**Table 4.** The summary of detected dominant signal information based on the direct capillary sequencing and cloning (identity %).

| Samples. | 18s rRNA | 18s rRNA; V4 | 16S rRNA; ITS | 16S rDNA | 18S rRNA |
|---|---|---|---|---|---|
| Non-treated | *Chlamydomonas nivalis* (99%) *Vitreochlamys nekrassovii* (99%) | *Aulacoseira granulate* (100%) *Aulacoseira ambigua*(99%) | *Chlorophyta sp.* (98%) | *Bacillus* cereus (81%) | *Tintinnidium fluviatile* (90%) |
| Treated | Not-detected | *Choanoflagellate* (97%) *Meira nashicola* (99%) | Not-detected | *Bacillus thuringiensis* (85%) | Not-detected |

\* The eukaryote universal primer, which was used to detect HNF (heterotrophic nanoflagellates), has detected Ciliophora (*Tintinnidium* sp.; 88%).

On the other hand, food sources identified in non-treated water sample, except for bacteria were not detected after the treatment of 2.5% diluted commercial bleach for 210 s, indicating their DNA fragments were eliminated by our selected treatment method (Figure 4A–E, T1~T2). Based on these results, we treated same process on *Asplanchna* specimen sorted from the reservoir for verifying if this pretreatment is proper to apply to rotifer species. Most identified species in non-treated water sample were not detected as gut contents in *Asplanchna*. However, Choanoflagellates, fungi species (*Meira*

sp.), and bacteria species (*Bacillus* sp.) were detected in some individuals. These species identified in treated *Asplanchna* samples seemed to have been detected by eliminating the signals that were strongly captured from the various DNA fragments that existed before the pretreatment. It means that selected commercial bleach as a pretreatment chemical and specified its concentration and exposure time properly can facilitate the removal of extracellular DNA fragments simultaneously with preserving rotifer body tissue, and consequently this process can be applied for detecting DNA of rotifers gut contents without fear of extracellular DNA contamination (Figure. 4A–E, A1~A3).

Unfortunately, it was difficult to interpret the detected band of bacteria in *Asplanchna* specimen as gut contents, because bacteria were not completely eliminated by the treatment of 2.5% diluted commercial bleach. The sequences from detected bands in the electrophoresis gel let us know that bacteria in non-treated and treated samples are species included in genus *Bacillus* (Table 4; Treated *Asplanchna* sample).

## 4. Discussion

As a chemical for pretreatment to remove external DNA on the lorica, we selected commercial bleach, which can be used on samples through the proper combination of bleach concentration and exposure duration time (seconds). Commercial bleach has been used mainly to prevent or eliminate contamination in DNA analyses (Table 1), but at the same time, it affects the body tissue of zooplankton, which can lead to the disintegration of rotifer loricas, and thus the release of rotifers' gut contents [33,35]. In this study, it was found that each rotifer species showed different response times when treated with commercial bleach at the same concentration, and the duration time for lorica survival differed by its characteristics (Table 3); the shortest time was for *Asplanchna* sp., having the softest lorica and the longest time for *Trichocerca* sp. and *Brachionus* sp. having lorica that are not easily damaged [49]. In the case of the genus *Keratella*, although its lorica has been suggested as a hard cover, which can be protective against mechanical interference by daphnids and predation by invertebrate predators [36], the loss of inner contents occurred through the mouth parts and not through lorica disintegration following the commercial bleach treatment. Therefore, based on the response time observed for *Asplanchna* sp. showing the shortest response time for disintegration after treatment with 2.5% diluted commercial bleach, we set the pretreatment time to 210 s, as this is an appropriate standard pretreatment method universally applicable to all rotifer taxa.

After applying this pretreatment method to raw water samples from the reservoir, it was confirmed that DNA fragments of rotifer food sources detected in non-treated samples were completely removed; Chlorophyceae, Diatomea, Cyanobacteria, and Ciliophora, except for bacteria (Figure 3, N, T1, T2). Further sequencing analyses indicated that the bacteria detected were mainly *Bacillus* sp. which is known to be tolerant and survive various removal treatments such as disinfection [50] (Table 4). *Bacillus* sp., gram-positive bacterium, has commonly been found in soil and other environments. It has been reported that *Bacillus* plays important roles in the lysis of bloom-forming blue-green alga and the control of their biomass in aquatic ecosystems [51,52]. Therefore, when we applied the pretreatment determined from this study to DNA analysis of gut content of rotifers, we cannot distinguish the source origins of bacteria detected in rotifer species, whether they came from contamination, water samples, or rotifer gut content, like the results of *Asplanchna* (Figure. 4D, A1~3). Since bacteria is one possible main food source for rotifers [53], a suitable pretreatment method for eliminating extracellular bacterial DNA should additionally be developed.

In the results based on DNA analysis, we used each group-specific primer for detection of targeted groups to confirm their presence/absence. As far as we know, there is no information about HNF-specific primer [54], so we applied instead a universal primer set for eukaryotes (Euk-A and Euk-B) which has been used to detect HNF (Table 2). We, therefore, carried out an additional experiment to define the applicability of the HNF primers set to rotifers. As a result, this primer set amplified all possible rotifer species from our study site except for *Asplanchna* sp. (Figure A1). These results provide a proper explanation for why the primer set did not work for all our samples (Figure 4F). The usage

of the primer sets that act specifically for each targeted biological community can help in improving detection accuracy for a targeted species group. However, there remain some limitations in verifying the effectiveness of a determined pretreatment on biological communities where specific primers have not yet been developed, such as HNF. In spite of these limitations, the results of the applied pretreatment method to *Asplanchna* sp. showed that specific food sources were detected in the gut content. Choanoflagellate, HNF species, has the habitat selection characteristic of being attached to phytoplankton species, and consequently it is expected that rotifers can eat Choanoflagellate indirectly in the process of eating phytoplankton, or select it as their food source directly [55]. In the case of *Meira nashicola*, which is a kind of yeast-like fungi species, although whether or not *M. nashicola* exists in aquatic ecosystem needs further research, it is considered a valid result of Asplanchna gut contents because parasitic fungus on phytoplankton, such as cyanobacteria, are known to feed on rotifers as alternative food sources [56]. So, when the limitations related to the detection of bacteria and HNF will be resolved, rotifer gut contents can be analyzed by pretreating with alcohol and commercial bleach as we recommend in the present study. Our study used traditional primer sets information; however, Adl et al. (2019) [54] recently revised the classification and nomenclature of Eurkaryotes and recommend some primer sets (rbcL, 18S V4) for Diatomea and Ciliophora. Therefore, we should apply these primer sets according to this new system for further study.

The main goal of the present study was to develop a pretreatment process that eliminated extracellular DNA fragments adhering to the Rotifera lorica and employing DNA barcoding, in order to accurately identify rotifer gut contents, thereby providing a better understanding of rotifer feeding behavior. We devised an experimental design for rotifer gut content analysis on the basis of DNA technology (DNA barcoding) while hypothesizing that feeding behavior (food selectivity) of rotifers with species-specific masticatory apparatus, e.g., the trophi, is dependent on the trophi characteristics. In this process of developing an experimental design, a pretreatment process for removing extracellular DNA as well as the cells attached to the rotifer lorica is essential in isolating accurately the DNA of the food sources remaining within each rotifer gut. Therefore, we selected appropriate chemicals for pretreatment and tried to establish the proper treatment bleach concentration (%) and duration time (seconds) by observing the response time for different types of lorica firstly fixed by 60% diluted ethanol and secondly treated with 2.5% diluted commercial bleach for 210 s. The final pretreatment process was tested on a water sample and a rotifer species (*Asplanchna* sp.) to verify its effectiveness. We conclude that the pretreatment process for rotifer worked effectively in removing extracellular DNA while enabling identification of selected food source taxa of rotifers using DNA barcoding. In this study, single PCR products from group-specific primers and the general eukaryotic primers for HNF were sequenced by the cloning and Sanger method. In forthcoming studies, the taxonomic diversity of the gut content may be analyzed using next-generation sequencing (NGS) while applying improved methods for the decontamination and selection of primers in the controlled experimental environments. The DNA analysis process of rotifer gut contents, especially the pretreatment process, can allow various approaches for DNA analyses for microinvertebrates whose feeding behavior is not sufficiently understood.

**Supplementary Materials:** The following are available online at http://www.mdpi.com/2076-3417/10/3/1064/s1. DNA sequences: Supplementary_Raw Sequences.FASTA (text format)

**Author Contributions:** conceptualization, H.-J.O., H.-G.J., and K.-H.C.; methodology, H.J., J.-S.G., K.-H.C. and H.J.; formal analysis, H.J., G.-J.J., S.-J.H. and H.J.; investigation, H.-J.O. and K.-H.C.; writing—original draft preparation, H.-J.O. and K.-H.C.; writing—review and editing, P.H.K., K.-H.C. and H.J.; visualization, H.-J.O., H.-G.J., and K.-H.C.; supervision, K.-H.C. and H.J.; project administration, I.-S.K.; funding acquisition, I.-S.K. and H.J. All authors have read and agreed to the published version of the manuscript.

**Funding:** This research was funded by the National Research Foundation of Korea, grant number NRF-2018R1A6A1A03024314.

**Conflicts of Interest:** The authors declare no conflict of interest.

## Appendix A

**Table A1.** Detailed previous treatment procedures in Table 1 (summary of previous treatment processes for decontamination in DNA analyses).

| Treatments | Procedure | Ref. |
|---|---|---|
| Psoralen + UV irradiation | 1. 8-methoxypsoralen of 100 μg·mL⁻¹<br>2. Irradiation with long-wave (365 nm) UV light for 1 h | [25] |
| Hydrogen peroxide<br>+ Bleach * + UV irradiation | 1. Soaked in hydrogen peroxide (3–30%) for 10–30 min<br>2. Rinsed with distilled water<br>3 *. Rinsed thoroughly with 10% bleach<br>4. Rinsed with distilled water<br>5. UV irradiated for 10 min | [26] |
| Acid wash + Ethanol<br>+ UV irradiation | 1. Soaked in 15% HCl for 10 min<br>2. Rinsed with 70% ethanol for 10 min<br>3. Rinsed in sterile double-distilled water for 30 min<br>4. UV irradiation (254 nm) for 15 min | [27] |
| Bleach * + Ethanol | 1 *. Soaked in 10% bleach for ~10 min<br>2. Rinsed with 70% ethanol | [28] |
| Bleach * + EDTA | 1 *. Immersed in 20% bleach for 2 min<br>2. Rinsed with distilled water<br>1. 0.5M EDTA at 55°C in a 2-day | [29] |
| UV irradiation + Bleach * | 1. UV irradiation (254 nm) for 10 min<br>2 *. Soaked for approximately 5 min in a 5% bleach solution | [30] |
| Acid wash + Bleach *<br>+ UV irradiation | 1. 30% acetic acid<br>2. Rinsed with ultrapure water<br>3 *. Immersed for 10 min in 10% sodium hypochlorite with sporadic shaking<br>4. Exposed to UV irradiation (254 nm) for at least 10 min | [31] |
| Ethidium Monoazied (EMA)<br>or<br>Propidium Monoazied (PMA) | 1. EMA or PMA added following a conventional procedure in accordance with the manufacturer's protocol (PowerSoil DNA extraction kit (Mo Bio Lab, Inc.) | [32] |
| Bleach * | 1 *. Exposure to 2.5% bleach for 40 min or overnight | [33] |

* Underlined treatment and procedure: the case of the application of bleach for decontamination

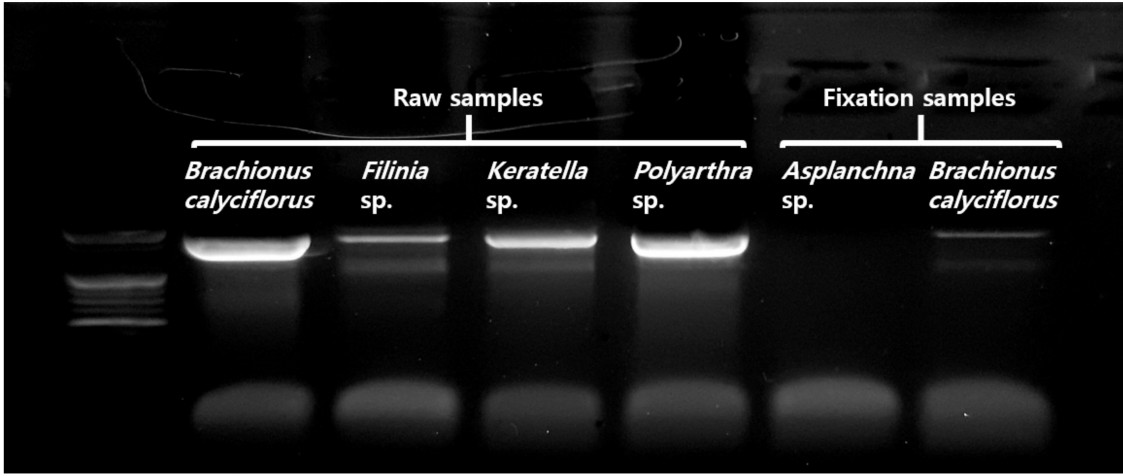

**Figure A1.** Results of applying Euk universal eukaryotic primers on each rotifer species (Raw samples: collected in the fresh water samples in the Shin-gal reservoir, fixation samples: stored in the laboratory).

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
