# Peer review of "Pretreatment Method for DNA Barcoding to Analyze Gut Contents of Rotifers"

_applsci, doi:10.3390/app10031064_

Round 1

Reviewer 1 Report

I think the authors have made some revisions that improve the manuscript. There remains some grammar issues although not many. These will be probably be corrected in preparation of proofs. 

Line 254, i don't think these are considered classes anymore. Maybe see a recent review and synthesis of taxonomic, nomenclature, and systematics changes in this paper "Journal of Eukaryotic Microbiology 2019, 66, 4–119". It would be good to use modern terms. 

Furthermore, the paper above also includes primers that are recommended. As the HF and HNF occur in many unrelated lineages, it will not be possible to develop HF specific primers, as it is a size category or a functional category, but not a phylogenetic lineage. Did the authors consult these tables and modern synthesis documents? 

Otherwise, i think the manuscript is fine as revised. 

Author Response

Reviewer 1.

I think the authors have made some revisions that improve the manuscript. There remains some grammar issues although not many. These will be probably be corrected in preparation of proofs. 

We appreciate your kind consideration of our revised paper (applsci-694479). We tried to do our best to revise our paper according to your comments. These changes are marked in RED.

Line 254, i don't think these are considered classes anymore. Maybe see a recent review and synthesis of taxonomic, nomenclature, and systematics changes in this paper "Journal of Eukaryotic Microbiology 2019, 66, 4–119". It would be good to use modern terms. 

We appreciate your great suggested reference. We revised taxonomic, nomenclature and systematics changes in our paper at line 25, 26, 152, 153, 256, 280, 286, 308.

Furthermore, the paper above also includes primers that are recommended. As the HF and HNF occur in many unrelated lineages, it will not be possible to develop HF specific primers, as it is a size category or a functional category, but not a phylogenetic lineage. Did the authors consult these tables and modern synthesis documents? 

We agree with your comment. Therefore, we reviewed above paper and discussed it at discussion part. Please see the line 320-321 and 340-343.

Otherwise, i think the manuscript is fine as revised. 

Again, we appreciate your kind consideration of our paper.

Reviewer 2 Report

In general, the study is useful. Especially for the subsequent use of next-generation sequencing to study the gut contents of such small organisms.

There are comments:

I could not find the DNA sequences (link did not work correct). It’s better to put sequences in GenBank or at least present it in FASTA text format in Supplementary Materials. In general, DNA sequencing and analysis of the obtained sequences is the weakest part of the manuscript.  It would be correct to provide more data on this section.

Line 89-90: insert reference

Line 114-115: insert reference about «typical concentration»

Line 178: insert reference to table 2

Line 191: better to write GenBank

Line 189-191: incorrect sentences

Figure 3: It is not clear why authors came to the conclusion that this is «extracellular DNA fragments»

Author Response

Reviewer 2.

In general, the study is useful. Especially for the subsequent use of next-generation sequencing to study the gut contents of such small organisms.

We appreciate your kind consideration of our revised paper (applsci-694479). We tried to do our best to revise our paper according to your comments. These changes are marked in RED.

There are comments:

I could not find the DNA sequences (link did not work correct). It’s better to put sequences in GenBank or at least present it in FASTA text format in Supplementary Materials. In general, DNA sequencing and analysis of the obtained sequences is the weakest part of the manuscript.  It would be correct to provide more data on this section.

We agree with your comment. We provide FASTA text format in Supplementary Materials. Please see the attached file.

Line 89-90: insert reference

We inserted relevant reference [24].

Line 114-115: insert reference about «typical concentration»

Thank you for your comment. We inserted reference [34] about typical concentration.

Line 178: insert reference to table 2

We inserted references. Please see the table 2.

Line 191: better to write GenBank

We changed word GENBANK to GenBank (line 190).

Line 189-191: incorrect sentences

We revised this incorrect sentences. Please see the revised sentences (line 189-190).

Figure 3: It is not clear why authors came to the conclusion that this is «extracellular DNA fragments»

We agree with your comment, therefore, we revised Figure 2. Please see new Figure 3 at line 245-247.

This manuscript is a resubmission of an earlier submission. The following is a list of the peer review reports and author responses from that submission.

Round 1

Reviewer 1 Report

The manuscript is not written in clear English and it makes many critical sentences or methodological passages cryptic.

There were too many sentences with incomprehensible meaning to list.

The keywords need to be redone professionally.

Lines 42-44. The presentation of the food web is erroneous. Rotifers, as invertebrate consumers, are a critical link between the microscopic and macroscopic trophic positions, but these are not separate or distinct food webs as the authors write. The presentation of the structure of the food web should be modernized, to reflect current thinking on the structure and function of food webs as a whole.

Line 11, soft illoricate, and gut-external DNA, these need to be rewritten in correct English.

Line 113, ethanol is the correct word.

Figure 1, legend is too brief and a long explanation is required. The two last depictions are not easy to understand and there is no explanation of the symbols and drawing. What’s going on in these?

The methods seem to indicate there were no replications. See line 146. This type of study requires multiple independent replications.

The NGS DNA techniques are notoriously difficult to replicate from environmental samples. That this study relies on a single run is problematic, and makes the analysis of data unreliable.

Protozoa and phytoplankton should be replaced by protist to reflect the reality of their trophic functional guilds. See Adl et al 2019 for a guide to understanding the vocabulary and the classification.

The manuscript should be clear about whether there were no replications, or what was replicated. If as indicated there were no replications, then this is not a piece of work that can be published, as it is unreliable.

The text repeats itself; there are sections of the discussion that repeat sections in the introduction, and from the results.

The discussion could be reduced by half. It is wordy and longwinded.

Overall the results ignore most of the work that has been done on removing environmental DNA. This study does not really provide any progress or new concept, or new technique.

I don’t believe this paper provides any advance, and it is not written clearly, and lacks adequate replications.

Author Response

Reviewer 1.

First of all, we appreciate your comments for our paper. We tried to do our best to revise our paper according to your comments and also carried out additional experiment (Figure 3). Please see the revised manuscript. Changes are marked in RED.

The manuscript is not written in clear English and it makes many critical sentences or methodological passages cryptic.

First draft version of this paper was proofread by commercial service (Enago Co.). We attached certification, please see the attached file. Revised draft was proofread by native Canadian (Associate Prof. Maurice Lineman, Taiyuan University of Technology).

There were too many sentences with incomprehensible meaning to list.

We did our best to improve incomprehensive sentences in the manuscript.

The keywords need to be redone professionally.

We revised keywords professionally.

Lines 42-44. The presentation of the food web is erroneous. Rotifers, as invertebrate consumers, are a critical link between the microscopic and macroscopic trophic positions, but these are not separate or distinct food webs as the authors write. The presentation of the structure of the food web should be modernized, to reflect current thinking on the structure and function of food webs as a whole.

We modified sentences according to the comment (line 41-43). We explained the role of rotifers focusing on the predators of micro-organisms and prey for invertebrate predators and fish. We deleted the expressions that separate the food webs.

Line 111, soft illoricate, and gut-external DNA, these need to be rewritten in correct English.

Thank you for your correction. We rewritten the words and sentences (line 111).

Line 113, ethanol is the correct word.

Thank you for your correction. We changed the word from ‘alcohol’ to ‘ethanol’ throughout the manuscript.

Figure 1, legend is too brief and a long explanation is required. The two last depictions are not easy to understand and there is no explanation of the symbols and drawing. What’s going on in these?

We agree with your comment. We modified figure and added detailed explanation of diagram. Please see the revised figure 1.

The methods seem to indicate there were no replications. See line 146. This type of study requires multiple independent replications.

There is some misunderstanding about replications for our experiment. We explain the replications at line 166 how many replicate we used. Water sample was used as a control for the experiment.

The NGS DNA techniques are notoriously difficult to replicate from environmental samples. That this study relies on a single run is problematic, and makes the analysis of data unreliable.

Our paper is preliminary study for pretreatment method for DNA analysis. We have plan to apply this method for ecological questions based on the NGS platform. Cost of commercial service for NGS platform has been decreased steeply. We can apply it to our further study.

Protozoa and phytoplankton should be replaced by protist to reflect the reality of their trophic functional guilds. See Adl et al 2019 for a guide to understanding the vocabulary and the classification.

The manuscript should be clear about whether there were no replications, or what was replicated. If as indicated there were no replications, then this is not a piece of work that can be published, as it is unreliable.

The text repeats itself; there are sections of the discussion that repeat sections in the introduction, and from the results.

The discussion could be reduced by half. It is wordy and longwinded.

We did our best to improve our paper throughout the manuscript according to your opinions.

Overall the results ignore most of the work that has been done on removing environmental DNA. This study does not really provide any progress or new concept, or new technique.

I don’t believe this paper provides any advance, and it is not written clearly, and lacks adequate replications.

We believe our paper has merits to understand rotifer’s feeding behavior and did our best to improve the paper. Please see the revised our paper.

Reviewer 2 Report

See attached file

Author Response

Reviewer 2.

Review for “Pretreatment method prior to rotifers gut contents analysis using DNA barcoding: response of rotifers to chemical treatment and application to elimination of extracellular DNA fragments in aquatic ecosystem”

The objective of this paper is to develop methods for isolating rotifer food sources from their guts, with the focus of distinguishing gut microorganisms from contamination (microorganisms from the surface of the animal and/or in the water samples).

This is an interesting topic, and a challenging project to pursue due to the relatively soft lorica of rotifers that prevents using methods established for crustaceans. However, there are several issues that need to be addressed before considering this manuscript for publication.

We appreciate your positive consideration of our paper. We tried to do our best to revise our paper according to your comments and also carried out additional experiment. Changes are marked in RED.

The paper, as written, lacks sufficient details in some of the methodological descriptions for the reader to fully understand what was done. The PCR and sequencing methods tend to bias the results towards very specific groups of microorganisms, and even then there are questions about the actually sequencing results (see below).

We revised method part to give a detailed information for understanding what was done sufficiently (fugure 1 and table 2). We selected primer sets for identify prey of rotifers based on Arndt (1993)’s review paper. To detect unexpected prey organisms, we have plan to apply university primer sets for eukaryotes with blocking primers.  

In addition, the experimental design seems to be lacking certain controls and the PCR/sequencing results themselves suggest a flaw with the molecular methods used (discussed in more detail below, with suggestions).

We used filtered water samples as a control which contain all possible prey organisms and added negative controls to prevent cross-contamination.

In addition, there are several instances of inadequate writing (grammatical errors, unclear writing, etc.) throughout the paper. If this manuscript is edited further and/or resubmitted, it is strongly suggested that the authors have the paper read and edited by someone proficient in English and familiar with biology/rotifers.

We carried out a English-proofreading it with native speaker who is Canadian and familiar with biology (Associate Prof. Maurice Lineman, Taiyuan University of Technology)

While the question is interesting, the methods used to address the question need to be developed better. In particular, why did the authors not expect the “HNF” PCR primers specific for eukaryotes to not amplify rotifer DNA? The fact that rotifer DNA was not amplified (as a positive control for the DNA isolation methods) suggests a flaw in the DNA isolation methods and/or PCR conditions/primers. In addition, the authors did not use PCR to test the samples after treating with bleach to determine if all external microorganisms had been removed (they could have isolated DNA and done PCR on the bleached sample after the rotifers were removed to ensure that all external microorganisms and DNA was removed by the bleach treatment), and negative controls were lacking for all PCRs.

We tried to find specific or group specific primer sets for heterotrophic nanoflagellates (HNF). However, we did not find adequate primer sets. Therefore, we used universal primer set to detect HNF according to reference [42, 43]. We have a plan to carry out the further study for developing adequate group specific primer sets based on this methodological study.

Therefore, it is suggested that the authors consider addressing the questions and suggestions below for improving this study before submission for publication. Below are comments and questions for the shortcomings of this manuscript and suggestions for improving the quality of the study before it could be considered for publication.

We appreciate your positive consideration of our paper. We tried to do our best to improve quality of our paper according to your comments.

Title: This is too wordy and should be simplified (e.g. “Using DNA barcoding to analyze feeding behavior of rotifers” or something much more straightforward.

We appreciate your great suggestion for the title of our study. I simplified the title based on your suggestion.

Introduction: The first three paragraphs attempt to justify the importance of studying rotifer feeding behaviors, but most of the citations (numbers 1-17) are not very recent (15 are prior to 2015). This implies that little research has been done studying rotifer feeding and DNA barcoding. If this is the case, then this should be explicitly stated. However, if more recent citations are available they should be included.

We updated references using recently published papers and add adequate references to support our paper. Please see the reference lists.

Lines 48-50: If rotifer-focused biological interactions (feeding behaviors) are receiving great attention, why are citations not provided here?

We provided reference to support our sentences (line 48-50, references number 7)

Line 69-70: Clarify what “their extraction” refers to.

We clarified what is their extraction. Please see the line 69-70.

Line 111: Should “illoricate” read “lorica” instead?

We replaced it.

Table 1 and Section 2.1: Table 1 seems to be a literature review of different methods that have been used with other organisms (although the organisms are not mentioned in the table. The authors then state on Line 110 that most of these methods were not suitable for rotifers, so it seems unnecessary to include all of this information. The authors then focus on using alcohol and bleach as their method of choice but sufficient methodological details are lacking. It would have been more informative to provide better protocol details (e.g. volumes used, timing, etc.) about the treatment of rotifers with alcohol and bleach in section 2.1 than the information in Table 1. It would be very difficult for someone to repeat what they did based on the information provided.

We revised Table 1 according to your suggestions. More detailed information moved to appendix I.

Line 91: Is the objective to isolate DNA fragments in the gut or organisms from the gut? Is the assumption that any microorganism present in the rotifer gut will be degraded to the point that only DNA fragments are present?

Our study focused on pretreatment method for eliminating extracellular DNA fragments of rotifers for DNA barcoding analysis and additionally tried to apply previously developed primer sets to identify potential prey organism groups. We will carry out the further research to improve our methods based on your comments.

Section 2.2: How were rotifer genera identified? In this section, reference is made to Figure 1, but several things in Figure 1 need to be explained better. For example:

What is the “treatment” referring to? Alcohol and bleach? Or one and/or the other? What does the “X” over Keratella mean? What do the other “X’s” mean? What are the curved arrows and small circles coming out from Asplanchna refer to? The description below the figure is very vague and not informative. We revised Figure 1 according to your comments. We clarified ambiguous expressions and added informative description of the figure legend. Please see the figure 1.

Section 2.3: Sampling methods are not explained well. The authors concentrated the water sample down to 1 liter, and then hand-removed organisms. Are these organisms the ones that comprise the 0.5ml of subsample referred to on Line 142? Was there only one 0.5 ml sample made from the 10L sample? Or was each 0.5ml derived from an independent 10L sample? What was contained within the 0.5ml sample? Rotifers or all microorganisms?

We clarified ambiguous expression for the sampling method (line 146-168)

Lines 145-146: Why are these organisms specifically chosen as the targets? This seems to bias the methods for detecting microorganisms (see below).

We selected potential prey organisms of rotifers based on Arndt (1993)’s [9] review paper. To detect unexpected prey organisms, we have plan to apply university primer sets for eukaryotes with blocking primers.

Lines 148-150: How were the samples filtered to prevent contamination from the air, filtration apparatus or from the researchers? More detail is needed. If the same thing was done for both the treated and non-treated, then this could be summarized in one sentence rather than in two sentences.

We added the detailed information to prevent contamination at the line 125-127 and 151-152 and 164-168. We merged two overlapped two sentences (line 155-156)

Lines 151-159: Why was Asplanchna selected and how does this sampling methods differ from the above method (does the 0.5ml sample above not contain rotifers? Could individual Asplanchna rotifers have not been selected form the 0.5ml sample?)

We did not carry out targeted sampling for Asplanchna. We selected Asplanchna for our study due to the ecological importance of aquatic food web according to Oh et al 2017 [7] and relatively easy to control than other rotifer group. We used individual Asplanchna for experiment (line 166). We explained why we select Asplanchna for our experiment at line 195-233

Lines 158-159: What does it mean to say that DNA fragments were absorbed by the cellulose nitrate filter? This is confusing since the next section (2.4) discusses how DNA was isolated with the Qiagen kit.

“absorbed” is “filtered”

Section 2.4, Line 171: How is a universal eukaryote primer applied to HNF? Wouldn’t this primer amplify DNA from all eukaryotes present, including the rotifers? If that wasn’t the goal, could you not have designed your own primer pair that is specific for HNF?

We tried to find specific or group specific primer sets for heterotrophic nanoflagellates (HNF). However, we did not find adequate primer sets. Therefore, we use universal primer set to detect HNF according to reference [42, 43]. We have a plan to carry out the further study for developing adequate primers based on this methodological study.

Lines 173 – 178: How were PCR products extracted from the gel?

We used commercial product to extract PCR product (AccuPrep® PCR/Gel DNA Purification Kit (50 reactions) [K-3038]) from the gel. Please see the line 182-183.

Table 2: Should also indicate the region of the rRNA genes that the primers target to demonstrate that variable regions are present to allow for interspecific/intergeneric comparisons.

We revised Table 2 to provide more clear and adequate information for the readers (line 190-194).

Results section:

Lines 185-197: It is still unclear what method was used to treat the samples. What was the effect of alcohol (if it was used)?

We used alcohol for the fixation cells in the samples to remove contaminants from the lorica efficiently.

Table 3: The time required for different bleach concentrations to cause lorica disintegration are provided, and variation among species is evident. The 2.5% concentration was then selected for moving forward. However, this does not explain the effect of the bleach on the microorganisms that might be present in the sample or present on the rotifer surface? Also, how were the rotifers removed from the bleach after they were treated? If bleach were carried forward to the DNA isolation method this might have affected the quality of the DNA. This entire bleach treatment method needed to be explained in much better detail.

The 10% concentration of bleach is generally used to remove extraneous contaminants. Our study selected 2.5% which is lower than 10%. We believe 2.5% concentration will not affect the quality of the DNA. However it need to carry out further study for explaining the effect of the bleach on the micororganisms. We insert new figure to visualize our experiment (figure 2).

Section 3.1: It would be nice to see an example (picture) of how bleach affects the lorica (how does disintegration appear under the microscope?).

Thank you for your suggestion. We added figure to show an example of disintegration process. Please see the figure 2 (line 214-217).

Section 3.2: Contamination of the treated sample is big question in this section. To ensure that the bleach removed extracellular DNA from the treated sample, the authors should have done a DNA extraction and PCR on the solution (no rotifers present) following bleach treatment. This could have been done by washing the rotifers form the nitrate filter, removing rotifers from the wash solution, and then doing DNA extractions on the wash solution. This would determine if all external microorganisms had been removed (one would only expect a PCR product from the rotifer DNA sample) and that any PCR products in the treated samples were actually from inside the rotifer. If any microorganisms survived the bleach treatment or were picked up by contamination they would have been detected by PCR. To make this even stronger, negative controls (containing no microorganisms) should have been controlled to ensure bacteria were not being amplified from the lab, reagents, air, etc.

Contamination problem is the most important issue in the PCR based analysis. We explained how to prevent cross contamination at the method part. We carried out additional experiment to add negative control as you mentioned. Please see the figure 3.

For the PCR results, I would have assumed that the bands on the gel would represent a mixture of PCR products from a variety of microorganisms present, but the authors described single sequences being amplified from each PCR product; this implies that only one genus was amplified from each sample. This was a surprising result (which is not addressed in the discussion) and likely an underrepresentation of what was actually present in the gut. A more thorough approach would have been to use universal 16S or 18S primers (one pair for prokaryotes and one pair for eukaryotes) and amplified DNA from each sample. Then, the purified PCR product could have been sequenced using Illumina or a similar next-generation sequencing method. This would have given a better description and relative quantification of the microbial populations within the rotifer gut, rather than biasing the sampling towards a few select groups, or only targeting taxa that could survive the bleach treatment. The authors do mention this as a future approach (Line 340), but the data presented here do not provide compelling evidence that they have a reliable method for isolating gut microorganisms that could be used with next-gen sequencing.

We recognized aforementioned problems. We just used capillary sequencing to obtain dominant signal from the PCR product. When we apply NGS approach, we can obtain many difference types of species (OTUs) from the targeted group. Our study focused on pretreatment method for eliminating extracellular DNA fragments of rotifers for DNA barcoding analysis and additionally tried to apply previously developed primer sets to identify potential prey organism groups. We have further study plan to apply it to NGS platform.

Lastly, all DNA sequences should be made accessible or presented in a supplementary document.

We deposited DNA sequences in the DAYAD. We will provide deposited link address soon.

Line 218: Should read 2.5% bleach.

We collected it.

Figure 3. Several comments about the gel:

Gel images are very low quality. PCR products and the DNA ladder are barely visible, so it is difficult to determine which reactions are actually blank. Why didn't the HNF primers work on all samples? Shouldn’t they have amplified rotifer DNA? If they are eukaryotic specific then there should have been PCR products in all five lanes (since rotifer DNA should be present). The absence of a PCR product from rotifers suggests a major flaw with the DNA purification and/or PCR amplification. Why is there not a negative control included for each reaction? This is necessary to ensure that any amplifications are not he result of contamination during DNA isolation, PCR, etc. What is the difference between the Asplanchna samples? Are they individual rotifers or biological replicates? Why were replicates not then done for the water samples? We carried out additional experiment based on your comments and revised it. Please see the new figure 3.

Lines 274-276: The importance of using alcohol was never really addressed. Was alcohol necessary? They could have processed a sample with bleach but without alcohol to test the effects. The inclusion of alcohol in the procedure was never strongly justified.

We used alcohol for the fixation cells in the samples to remove contaminants from the lorica efficiently.

References

Arndt, H. Rotifers as predators on components of the microbial web (bacteria, heterotrophic flagellates, ciliates)—a review. In Rotifer Symposium VI.; Gilbert, J.J., Lubzens, E., Miracle, M.R., Eds.; Springer: Dordrecht, Nederland, 1993; pp. 231-246.

Reviewer 3 Report

The manuscript tackles a very interesting topic and provides several great ideas on how to address the issue of gut content analysis in microscopic animals.

I find the manuscript well written and surely deserving to be published, but I also found some weaknesses that need to be solved, or at least acknowledged in the manuscript.

Rotifers were sampled from the field, and were not fed with a specific group of organisms: thus, the results cannot be checked for reliability. For example, all the gut content obtained from DNA is from algae, whereas Asplanchna is mostly a predator. The reader needs to be convinced that the analysed Asplanchna indeed fed only on algae and not on other rotifers. Thus, a laboratory-fed population of Asplanchna should be used, with differential treatments of them: some replicates with only algae as food, some replicates with only small rotifers as food, some replciates with only ciliates as food, etc., and other replicates with combinations of prey items. Only such approach will provide convincing evidence that the method works identifying the preys that should be in the gut content of Asplanchna. The current results are not convincing. The DNA barcoding approach that was applied can amplify and sequence successfully only one DNA sequence at a time: if multiple sequences are there, only cloning could provide useful chromatograms to check whether the PCR product indeed refers to the targeted prey. The method of capillary sequencing could be OK, but an Illumina sequencing with metabarcoding could do a better job, using different primers to amplify shorter fragments. Moreover, the universal primers for eukaryotes could have sequenced also rotifer preys.

Minor issues:

The type of sample used for DNA extraction is not clear. The text on line 151-156 deals with single individuals of Asplanchna, but then from line 157 to 161, the term “sample” is used without clarifying whether each sample is composed of single rotifers or of bulk population of Asplanchna.

Some minor issues with the language need to be fixed. For example (but not only) “their soft illoricate”, “verificaetion”, “oconcentration”, etc.

Author Response

Reviewer 3.

The manuscript tackles a very interesting topic and provides several great ideas on how to address the issue of gut content analysis in microscopic animals.

I find the manuscript well written and surely deserving to be published, but I also found some weaknesses that need to be solved, or at least acknowledged in the manuscript.

We appreciate your kind consideration of our paper. We tried to do our best to revise our paper according to your comments and also carried out additional experiment. Changes are marked in RED.

Rotifers were sampled from the field, and were not fed with a specific group of organisms: thus, the results cannot be checked for reliability. For example, all the gut content obtained from DNA is from algae, whereas Asplanchna is mostly a predator. The reader needs to be convinced that the analysed Asplanchna indeed fed only on algae and not on other rotifers. Thus, a laboratory-fed population of Asplanchna should be used, with differential treatments of them: some replicates with only algae as food, some replicates with only small rotifers as food, some replciates with only ciliates as food, etc., and other replicates with combinations of prey items. Only such approach will provide convincing evidence that the method works identifying the preys that should be in the gut content of Asplanchna. The current results are not convincing. The DNA barcoding approach that was applied can amplify and sequence successfully only one DNA sequence at a time: if multiple sequences are there, only cloning could provide useful chromatograms to check whether the PCR product indeed refers to the targeted prey. The method of capillary sequencing could be OK, but an Illumina sequencing with metabarcoding could do a better job, using different primers to amplify shorter fragments. Moreover, the universal primers for eukaryotes could have sequenced also rotifer preys.

We appreciate your detailed comments to improve our paper. Our study focused on pretreatment method for eliminating extracellular DNA fragments of rotifers for DNA barcoding analysis and additionally tried to apply previously developed primer sets to identify potential prey organism groups. Therefore, we need to develop specific and adequate primer sets for the further study and also have a plan to carry out the further study according to your comments based on this methodological study. Our revised manuscript was almost rewritten by the authors according to your comments and also carried out additional experiment (figure 3). Please see the fully revised manuscript.

Minor issues:

The type of sample used for DNA extraction is not clear. The text on line 151-156 deals with single individuals of Asplanchna, but then from line 157 to 161, the term “sample” is used without clarifying whether each sample is composed of single rotifers or of bulk population of Asplanchna.

Thank you for your comment. We clarified expression at line 167-168.

Some minor issues with the language need to be fixed. For example (but not only) “their soft illoricate”, “verificaetion”, “oconcentration”, etc.

We revised words spelled wrongly throughout the manuscript.

Round 2

Reviewer 2 Report

See attached file

Author Response

Response to “Pretreatment method for DNA barcoding to analyze 2 gut contents of rotifers” The authors have put in a great deal of work to address many of the concerns expression in the previous review. In particular, some major improvements are:

The title is more concise and easier to understand. The description for Figure 1 is better. The justification for including Table 1 makes more sense now The inclusion of Figure 2 is a nice addition to the paper. The gel picture in Figure 3 (with the inclusion of a negative control) is much improved.

Many other responses to specific queries were addressed.

--> We appreciate your kind consideration of our revised paper (applsci-557586). We worked hard to account for all your comments in this 2nd round revision. Furthermore, we also performed one more additional experiment to explain the 1st issue which was also related to the 2nd issue. These changes are marked in RED.

However there are still two major issues with the manuscript that raise concerns:

The issue with the universal primers (which is a major focus of the paper) is still unresolved. The authors selected primers EukA and EukB, which are universal primers for eukaryotes (as stated in Table 2) yet they failed to amplify rotifer DNA (Figure 3). Why did the primers not work for all samples? Shouldn’t they have amplified rotifer DNA as a positive control? This suggests a potential flaw somewhere in the design. The authors need to explain why eukaryote-specific primers failed to amplify rotifer DNA. --> We agree with your comment. Therefore, we carried out an additional experiment to define an applicability of the HNF primers set to the rotifers using raw samples (collected in the fresh water samples in the Shin-gal reservoir [study site]) and fixation samples (stored in the laboratory). As a result, the primers set amplified all possible rotifer species which inhabited in our study site except Asplanchna sp. (Appendix II). It provide us a proper explanation why did not the primers set work for all our samples (Figure 4-(F)). We explained it at line 308-322. The DNA sequencing of the PCR products (how a DNA sequence was obtained from a PCR product that contains a mixed population of sequences) is still not clear to me. In the response, the authors’ state, “We just used capillary sequencing to obtain dominant signal from the PCR product.” Does this imply that a mixture of DNA sequences (from a single PCR product) was simultaneously sequenced in a single reaction, and then the dominant signal from the chromatogram was arbitrarily determined? If so, this would not be the correct way to do this. Since NGS is not feasible at this point, the authors should have considered cloning the PCR product and then sequencing a few clones to identify what sequences were present (they would have therefore sequenced individual sequences). Or do the authors this mean that there was only one species in each sample that was dominant? If this were the case, what is more surprising is that if the EukA and EukB primers are universal for eukaryotes, then the rotifer DNA sequence should have been the dominant signal. But this was not the case, so it is unclear how the sequencing was able to recover a clean chromatogram of a non-rotifer sequence. --> We appreciate your comments. We fixed the text for clarification. Please see the line 337-342.

These two points above need further explanation, since they are a central focus to the objectives of the paper.

Lastly, while the English/writing is improved (I appreciate the effort made by the authors to find a colleague to assist with this), there are still several sections that need attention, and I assume the editorial staff can assist with this if the paper were accepted for publication.

--> We had Prof. Paul Henning Krogh (Aarhus Univeristy, Denmark) and Maurice Lineman (Taiyuan University of Technology, China) read the article for linguistic and textual accuracy.

Reviewer 3 Report

The manuscript did not improve since its first submission. The methods are still not convincing and the experimental approach cannot answer the question. The changes performed on the text did strengthen the inference. I am sorry for my negative assessment, but I do not think that the story can be supported: a clearer and more structured rationale should be used to support the results and the reliability of the inference. This is a pity, because a lot of work was performed to obtain the data.

The English language did not improve either: there are some sentences that may be grammatically correct but that convey a wrong message, such as that fish are predators of bacteria in the first paragraph of the introduction. The manuscript has several such problems that need to be fixed.

Author Response

The manuscript did not improve since its first submission. The methods are still not convincing and the experimental approach cannot answer the question. The changes performed on the text did strengthen the inference. I am sorry for my negative assessment, but I do not think that the story can be supported: a clearer and more structured rationale should be used to support the results and the reliability of the inference. This is a pity, because a lot of work was performed to obtain the data.

--> First of all, we appreciate your comments for our paper. We tried to do our best to revise our paper according to your comments and carried out an additional experiment. We believe our paper has merits in enabling understanding of rotifer feeding behavior and did our best to improve the paper. We attached first and last version of our paper. Please see the revised our paper.

The English language did not improve either: there are some sentences that may be grammatically correct but that convey a wrong message, such as that fish are predators of bacteria in the first paragraph of the introduction. The manuscript has several such problems that need to be fixed.

--> We had Prof. Paul Henning Krogh (Aarhus Univeristy, Denmark) and Maurice Lineman (Taiyuan University of Technology, China) read the article for linguistic and textual accuracy.
